# Is Sentinel Lymph Node Biopsy for Breast Cancer with Cytology-Proven Axillary Metastasis Safe? A Prospective Single-Arm Study

**DOI:** 10.3390/jcm10204754

**Published:** 2021-10-16

**Authors:** Hee Jun Choi, Jai Min Ryu, Byung Joo Chae, Seok Jin Nam, Jonghan Yu, Se Kyung Lee, Jeong Eon Lee, Seok Won Kim

**Affiliations:** 1Department of Surgery, Samsung Changwon Hospital, Sungkyunkwan University School of Medicine, Changwon 51353, Korea; chj21899@gmail.com; 2Division of Breast Surgery, Department of Surgery, Samsung Medical Center, Sungkyunkwan University School of Medicine, Seoul 06351, Korea; jaimin.ryu@samsung.com (J.M.R.); bj.chae@samsung.com (B.J.C.); seokjin.nam@samsung.com (S.J.N.); jonghan.yu@samsung.com (J.Y.); sekyung.lee@samsung.com (S.K.L.); jeongeon.lee@samsung.com (J.E.L.)

**Keywords:** sentinel lymph node biopsy, axillary lymph node dissections, cytology-proven axillary metastasis

## Abstract

The purpose of this study was to evaluate pathologic lymph node metastasis in breast cancer with cytology-proven axillary metastasis. This study was designed prospectively. We performed axillary lymph node dissections (ALND) after lymphatic mapping by near-infrared (NIR) fluorescence imaging with Indocyanine Green (ICG). We evaluated 72 breast cancer patients with cytology-proven axillary metastasis by curative surgery at the Samsung Medical Center between May of 2016 and December of 2017. Among the 72 patients with cytology-proven axillary metastasis, 14 of 39 patients (35.9%) with one or two sentinel lymph nodes containing metastases were metastasized to post-sentinel lymph node. Thirteen of fourteen patients had additional non-sentinel lymph node metastases, seven of thirteen patients also had additional level II lymph node metastases, and one patient had only one additional level II lymph node metastasis. Of T1 or T2 stage patients, 10 of 33 patients (30.3%) with one or two sentinel lymph nodes containing metastases were metastasized to post-sentinel lymph node. Even in patients without SLN metastasis, 50% of the patients had at least three LN metastases, and 40% in the T1 or T2 stage patients. Sentinel lymph node biopsy without ALND might be not safe for patients with cytology-proven axillary metastasis.

## 1. Introduction

Nodal status is the primary prognostic indicator in breast cancer, and it is important for determining treatment [1]. Recent studies have found that axillary lymph node dissection (ALND) was unnecessary for patients with nonpalpable axillary lymph nodes (LN) with one or two positive sentinel lymph nodes (SLN) [2,3,4].

The American College of Surgeons Oncology Group (ACOSOG) did not conduct a preoperative axillary ultrasound (US) [3] in its Z0011 trial; however, in many medical centers, preoperative axillary US is standard procedure for breast cancer patients and axillary US-guided LN needle biopsy is implemented for suspicious LNs. Generally, ALND is required in breast cancer patients with cytology-proven axillary metastasis; however, a recent study demonstrated that a positive US-guided LN needle biopsy in breast cancer may be conducted with a sentinel lymph node biopsy (SLNB) [5].

If patients with cytology-proven axillary metastasis have one to two sentinel lymph node metastases and are omitted ALND, the existence of other axillary metastases cannot be confirmed. The purpose of this study was to evaluate pathologic lymph node metastasis in breast cancer with cytology-proven axillary metastases.

## 2. Methods

This study was prospective. We included 147 invasive breast cancer patients with proven pre-operation cytology and axillary LN metastasis followed by curative surgery at the Samsung Medical Center between May 2016 and December 2017. The inclusion criteria were patients with abnormal axillary US, axillary LN metastases confirmed preoperatively by US-guided fine needle aspiration, and patients who were planning to undergo ALND. Cases were excluded due to stage IV disease at presentation or for previous ipsilateral axillary surgery. We implemented fine needle aspiration of suspicious LN on axillary US; however, we did not investigate the number of suspicious LNs. We performed an axillary lymph node dissection (ALND) with lymphatic mapping using near-infrared NIR fluorescence imaging with ICG. ICG was used to find the SLNs. Twenty-two patients had ICG failure due to lymphatic obstruction and inflammation breast cancer, and 26 patients had an axillary pathologic complete response (pCR) with neoadjuvant chemotherapy (NAC). We excluded an additional 27 patients with NAC. We finally evaluated 72 breast cancer patients with cytology-proven axillary metastases without NAC. Additionally, we evaluated 60 patients with T1 or T2 stages of these 72 patients.

A total of 10 cc of ICG diluted 121 times was prepared. A volume of 1 cc of diluted ICG was injected intradermally and subcutaneously into the breast. The patient underwent ALND after injection, and lymph node dissection was performed using a NIR fluorescence camera. First, we removed the fluorescence lymph node, which was considered the SLN. Then, we removed the remaining axillary lymph nodes in axillary levels I and II. We obtained the results of axillary lymph node by classification (Figure 1). We measured, froze, and serially sectioned the excised SLNs transversely into 16 or 24 slices. After pathological evaluation of the sections, we fixed the remaining tissue in 10% formalin, embedded it in paraffin blocks, and finally prepared hematoxylin and eosin (H&E)-stained sections. We designated metastatic foci of 0.2 to 2 mm as micrometastases, and metastatic clusters smaller than 0.2 mm were considered isolated tumor cells, whether they were detected by H&E or immunohistochemistry.

In this study, axillary level I was defined as the bottom level, below the lower edge of pectoralis minor muscle and axillary level II was defined as lying underneath the pectoralis minor muscle. Post-SLN metastasis was defined as non-SLN metastasis or level II LN metastasis after SLN metastasis. We distinguished SLN metastasis with fluorescence and post-SLN metastasis. We categorized patients in whom SLNs were identified as true positive (TP) or false negative (FN). The false negative rate (FNR) was calculated by FN/(FN + TP). We worked with the statistics team at Samsung Medical Center to calculate the FNR. This study was approved by the Institutional Review Board of Samsung Medical Center, Seoul, Korea (IRB file no. 2015-01-046). All patients provided written informed consent and the Clinical trials.gov identifier is NCT02781259.

## 3. Results

We performed lymphatic mapping for 72 patients with cytology-proven axillary metastasis using ICG. The demographic and clinicopathological characteristics of the patients included in this study are listed in Table 1. The identification of SLN with ICG was 100% (72 of 72). The FNR of SLNB with cytology-proven axillary metastasis by ICG was 8.3% (6 of 72) in all patients and 8.3% (5 of 60) in T1 or T2 patients. There were no patients with only micrometastasis. Among the 72 patients, 6 had no SLN metastasis, 39 had one to two SLN metastases, and 27 had more than three or more SLN metastasis. Among the 6 patients without SLN metastasis, 5 patients had N1 stage and 1 had N2 stage. Three patients of five with N1 stage had two LN metastases, and 2 of 5 had three LN metastases. Thus, 3 patients (50%) without SLN metastasis had more than three LN metastases. Among the 39 patients with one to two SLN metastases, 14 (35.9%) were metastasized to post-SLN. Thirteen patients had additional non-SLN metastasis, and 7 of 13 patients also had additional level II lymph node metastases. One patient had only an additional level II lymph node metastasis. Twenty-seven patients had three or more sentinel lymph node metastases, and post-SLN metastasis occurred in 17 (63.0%) patients (Figure 2).

Sixty patients of seventy-two had T1 or T2 on tumor stage. There were 5 patients without SLN metastasis, 33 patients with one to two SLN metastases and 22 patients with more than 3 sentinel lymph node metastases. Among 5 patients without sentinel lymph node metastasis, 2 patients (40%) had more than three LN metastases. Among the 33 patients with one to two SLN metastases, 10 (30.3%) were metastasized to post-SLN. Nine patients had additional non-sentinel lymph node metastasis, and 4 of 9 patients also had additional level II lymph node metastases. One patient had only an additional level II lymph node metastasis (Figure 3).

## 4. Discussion

In this prospective study, we surveyed lymphatic metastasis of breast cancer in pre-operative cytology-proven axillary metastasis. ICG was used for identifying SLNs and non-SLNs. Previous reports using ICG for breast cancer showed that the identification rate of SLN was 92–100% [6,7,8,9,10,11]. Because isolating SLN using ICG is very reliable, feasible and does not involve exposure to radiation, we used ICG to identify SLNs during ALND.

A recent study demonstrated that 73% of patients without palpable lymphadenopathy but with preoperative US-guided biopsy proven axillary LN metastases had N1 disease. This finding suggests that patients with a tumor size ≤2 cm and one abnormal LN on axillary US may undergo SLNB [5]. Our study showed that 16 patients had tumor sizes ≤2 cm, and, of those patients, 12 patients had N1 disease. However, four (33.3%) of 12 patients had three LN metastases.

Pohlodek et al. studied prediction of additional lymph node involvement in breast cancer patients with positive SLN. This study told only tumor stage was significant in predicting the metastasis in non-SLN with positive SLN [12]. In our study, 35.9% of all patients with one to two SLN metastases had metastasis to post-SLN, and 30.3% of T1 or T2 stage metastases to post-SLN.

In the recent ACOSOG Z0011 trial, among patients with one or two sentinel lymph nodes containing metastases, the 10-year disease-free survival rate was 80.2% in the group with only SLNB, and was 78.2% in the ALND group. Furthermore, the 10-year overall survival was 86.3% in the group with only SLNB, and 83.6% in the ALND group [3]. A radiotherapy or surgery (AMAROS) trial determined that SLN biopsy without ALND offered excellent regional control, and this regimen may be a reasonable option for disease management [2]. Although our study involved patients who were axillary lymph node positive before surgery, among the patients with one or two sentinel lymph nodes containing metastases, 14 of 39 patients (35.9%) metastasized to post-sentinel lymph node. Thirteen of 14 patients had additional non-sentinel lymph node metastases, seven of 13 patients also had additional level II lymph node metastases, and one patient had only one additional level II lymph node metastasis. In T1 or T2 stage disease, 10 patients (30.3%) were metastasized to post-sentinel lymph node. Even in patients without SLN metastasis, 50% of the patients had at least three LN metastases, and 40% in the T1 or T2 stage. Patients that were axillary lymph node positive before surgery might also have additional axillary LN metastasis, therefore SLNB could not be a safe choice because of the risk of undertreatment for patients with high nodal burden.

This study had some limitations. First, it was limited to a single comprehensive cancer institution. Second, the study did not exclude patients based on clinical T stage, and therefore it was difficult to compare our patients to those in the Z0011 trial. Third, this study did not examine oncological outcomes, but investigated additional lymphatic metastases. Fourth, this study used a small group of patients. As such, it was difficult to provide reliable statistics from our findings. However, this study is meaningful even if the number of patients was relatively small because it was prospective, and did not investigate clinically suspected node-positive patients, but node-positive patients confirmed by cytology.

In conclusion, it may not be safe to omit ALND in patients who are cytology axillary node-positive when zero to two SLN metastases are found. Further studies are needed with larger groups of patients.

## Figures and Tables

**Figure 1 jcm-10-04754-f001:**
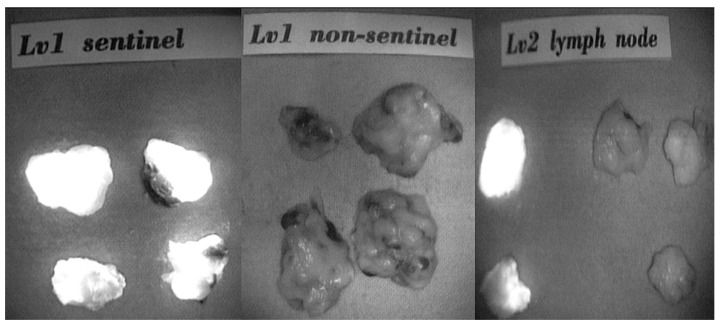
Axillary lymph node obtained by classification (sentinel node, non-sentinel node, and level II node).

**Figure 2 jcm-10-04754-f002:**
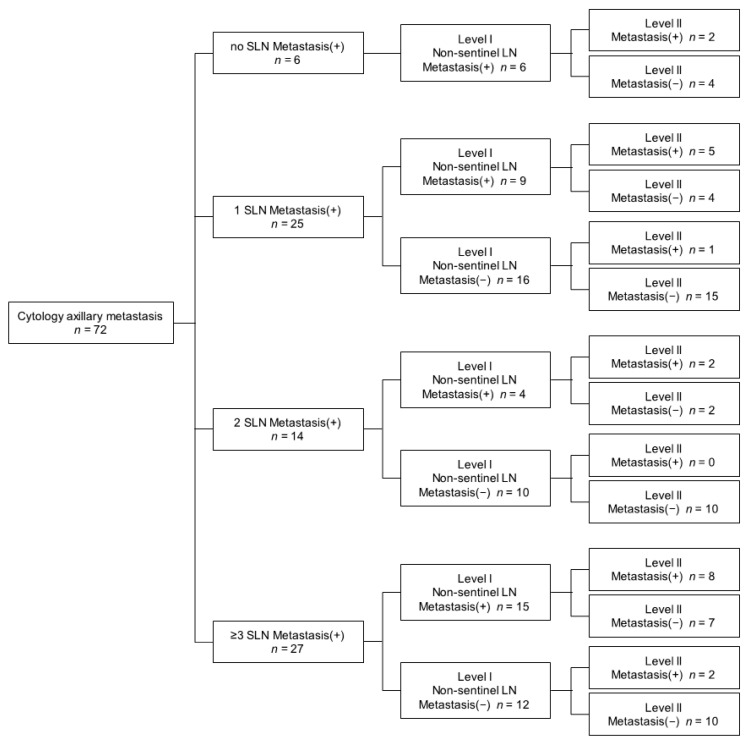
Algorithm of lymph node metastasis in breast cancer with cytology-proven axillary metastasis.

**Figure 3 jcm-10-04754-f003:**
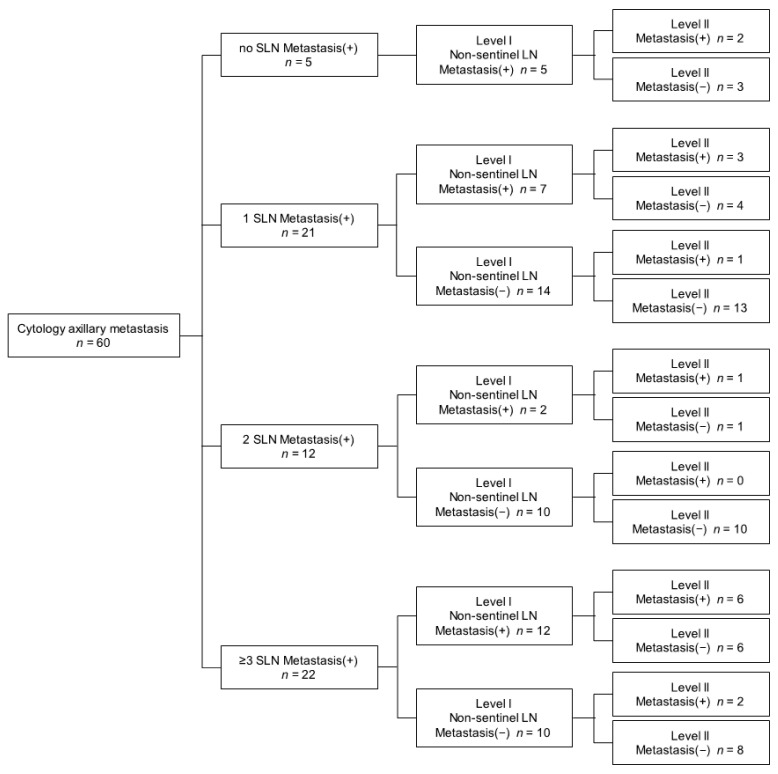
Algorithm of lymph node metastasis in breast cancer with cytology-proven axillary metastasis with T1 to T2 stage.

**Table 1 jcm-10-04754-t001:** Clinicopathological and characteristics of patients.

Characteristic	No	%
Age, years		
Median	49.2 ± 9.3	
Range	(25–66)	
Pathologic T stage		
T1	16	22.2
T2	44	61.1
T3	12	16.7
Pathologic N stage		
N1	37	51.4
N2	23	31.9
N3	12	17.7
Clinical tumor subtype		
ER positive and/or PR positive, HER2 negative ER positive and/or PR positive, HER2 positive	64 6	88.9 8.3
HER2 positive	1	1.4
Triple negative	1	1.4
Lymphovascular invasion	36	50.0
No SLN metastasis	6	8.3
1 to 2 SLN metastasis	39	54.2
Type of surgery		
Mastectomy	33	45.8
Breast conserving surgery	39	54.2
Total	72	100.0

*SLN*—sentinel lymph node, *ER*—estrogen receptor, *PR*—progesterone receptor, *HER2*—human epidermal growth factor 2.

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
