# Peer review of "Is Sentinel Lymph Node Biopsy for Breast Cancer with Cytology-Proven Axillary Metastasis Safe? A Prospective Single-Arm Study"

_jcm, 2021, doi:10.3390/jcm10204754_

Round 1

Reviewer 1 Report

I like this paper and it is on a relevant and important clinical topic.  The content is fine but it is written poorly and needs clear and full editing before it is ready for publication.  Some of the sentences are written in a way that confuses the data.

Author Response

I like this paper and it is on a relevant and important clinical topic.  The content is fine but it is written poorly and needs clear and full editing before it is ready for publication.  Some of the sentences are written in a way that confuses the data.

Answer> Thank you good comments. As the review recommended, we had English full editing and corrected some sentences in revised manuscript. If this is not enough, we will make additional corrections.

Reviewer 2 Report

This study was aiming to evaluate pathologic lymph node metastasis in breast cancer with cytology-proven axillary metastasis prospectively. One major contribution of this work is to suggest that it may be not safe for these patients to omit ALND when 0-2 positive SLNs are found preoperatively. The comments below may be considered.

  1. In the Introduction section, Line 41-42, “If patients with cytology-proven axillary metastasis have one to two sentinel lymph node metastases, the existence of other axillary metastases cannot be confirmed.” It would be easier to understand the intent of this part if you added a sentence such as "If ALND is omitted in these patients.
  2. Regarding the inclusion criteria, did all 72 patients in the study have nonpalpable axillary lymph nodes?
  3. Please describe the definitions of axillary level I and level II.
  4. Please describe the definitions of post-SLN. It is a little difficult to understand the difference between non-SLN and post-SLN from the text alone, although we can understand it by referring to the figures.
  5. In the Results section, the results of the evaluation of all 72 patients were described first, followed by the results of the evaluation of 60 patients with T1-T2 stage. However, in the Methods section, there was no description about performing such evaluations.
  6. In the Results section, the metastatic status of post-SLN was described, but in the Methods section, there was no description about performing such evaluations.
  7. In the Methods section, there was a description about metastatic size (i.e., ITC, micrometases, macrometastases), but in the Results section, there was no description about this point.
  8. In the Methods section, Line 75-76, “We used SPSS version 23 for the chi-square tests.” However, in the Results section, no description about statistical analysis was shown.
  9. In the Results section, Line 93-94, “Twenty-two patients had three or more sentinel lymph node metastases, and metastasis to post-SLN occurred in 14 (63.6%) patients (Fig.2)”. But, this was the result of the patients with T1-T2 stage, not the result of all 72 patients. I think the correct description is “Twenty-seven patients had three or more sentinel lymph node metastases, and metastasis to post-SLN occurred in 17 (63.0%) patients (Fig.2)”.
  10. In the Results section, Line 106, “The identification of SLN with ICG is 100% (72/72).” Please move this sentence to before “The FNR of SLNB with… .” (Line 83)
  11. In the Discussion section, the authors said “Patients that were axillary lymph node positive before surgery might also have additional axillary LN metastasis, therefore SLNB could not be safe choice at these patients.” However, in the ACOSOG Z0011 study, at least 30 % of patients in the SLNB group were thought to have residual metastatic lymph nodes. Therefore, it cannot be concluded that SLNB could not be a safe choice just because this study had showed that additional metastatic nodes were detected in the post-SLNs. In the Results section, Table 1 showed that patients with cytology-proven axillary metastasis had a higher percentage of heavy (≥4 positive lymph nodes) nodal disease burden (35/72, 48.6%). Wouldn't it be better to state that SLNB could not be a safe choice because of the risk of undertreatment for patients with high nodal burden?
  12. In relation to above mention, it is well known that the number of suspicious LNs on axillary US is an important predictor of LN disease burden. Please describe whether this study investigate the number of suspicious LNs on axillary US.

Author Response

This study was aiming to evaluate pathologic lymph node metastasis in breast cancer with cytology-proven axillary metastasis prospectively. One major contribution of this work is to suggest that it may be not safe for these patients to omit ALND when 0-2 positive SLNs are found preoperatively. The comments below may be considered.

  1. In the Introduction section, Line 41-42, “If patients with cytology-proven axillary metastasis have one to two sentinel lymph node metastases, the existence of other axillary metastases cannot be confirmed.” It would be easier to understand the intent of this part if you added a sentence such as "If ALND is omitted in these patients.

Answer> Thank you good comments. As the review recommended, we added the sentence in the introduction.

  1. Regarding the inclusion criteria, did all 72 patients in the study have nonpalpable axillary lymph nodes?

Answer> Thank you good comments. 72 patients in the study have palpable or nonpalpable axillary lymph nodes. Patients were included abnormal axillary US, confirmed axillary metastasis by FNA preoperatively and planed ALND. We revised Method in the revised manuscript.

  1. Please describe the definitions of axillary level I and level II.

Answer> Thank you good comments. As the review recommended, we added the definition of axillary level I and II in the Method.

  1. Please describe the definitions of post-SLN. It is a little difficult to understand the difference between non-SLN and post-SLN from the text alone, although we can understand it by referring to the figures.

Answer> Thank you good comments. As the review recommended, we added the definition of post-SLN metastasis in the Method.

  1. In the Results section, the results of the evaluation of all 72 patients were described first, followed by the results of the evaluation of 60 patients with T1-T2 stage. However, in the Methods section, there was no description about performing such evaluations.

Answer> Thank you good comments. As the review recommended, we added evaluation of 60 patients with T1-T2 stage in the Method.

  1. In the Results section, the metastatic status of post-SLN was described, but in the Methods section, there was no description about performing such evaluations.

Answer> Thank you good comments. As the review recommended, we added the description about performing such evaluation in the Method.

  1. In the Methods section, there was a description about metastatic size (i.e., ITC, micrometases, macrometastases), but in the Results section, there was no description about this point.

Answer> Thank you good comments. We did analysis about metastatic size, there was none patient with only micrometastasis. As the review recommended, we added the description in the Results.

  1. In the Methods section, Line 75-76, “We used SPSS version 23 for the chi-square tests.” However, in the Results section, no description about statistical analysis was shown.

Answer> Thank you good comments. We did this analysis, however it was not analysis needed for the results. So we revised Method in the revised manuscript.

  1. In the Results section, Line 93-94, “Twenty-two patients had three or more sentinel lymph node metastases, and metastasis to post-SLN occurred in 14 (63.6%) patients (Fig.2)”. But, this was the result of the patients with T1-T2 stage, not the result of all 72 patients. I think the correct description is “Twenty-seven patients had three or more sentinel lymph node metastases, and metastasis to post-SLN occurred in 17 (63.0%) patients (Fig.2)”.

Answer> Thank you good comments. We agree that twenty-seven patients had three or more sentinel lymph node metastases, and post-SLN metastasis occurred in 17 (63.0%) patients. As the review recommended, we revised results in the revised manuscipt.

  1. In the Results section, Line 106, “The identification of SLN with ICG is 100% (72/72).” Please move this sentence to before “The FNR of SLNB with… .” (Line 83)

Answer> Thank you good comments. As the review recommended, we moved the sentence and revised results in the revised manuscipt.

  1. In the Discussion section, the authors said “Patients that were axillary lymph node positive before surgery might also have additional axillary LN metastasis, therefore SLNB could not be safe choice at these patients.” However, in the ACOSOG Z0011 study, at least 30 % of patients in the SLNB group were thought to have residual metastatic lymph nodes. Therefore, it cannot be concluded that SLNB could not be a safe choice just because this study had showed that additional metastatic nodes were detected in the post-SLNs. In the Results section, Table 1 showed that patients with cytology-proven axillary metastasis had a higher percentage of heavy (≥4 positive lymph nodes) nodal disease burden (35/72, 48.6%). Wouldn't it be better to state that SLNB could not be a safe choice because of the risk of undertreatment for patients with high nodal burden?

Answer> Thank you good comments. As the review recommended, we agree that SLNB could not be a safe choice because of the risk of undertreatment for patients with high nodal burden. We revised Discussion in the revised manuscript.

  1. In relation to above mention, it is well known that the number of suspicious LNs on axillary US is an important predictor of LN disease burden. Please describe whether this study investigate the number of suspicious LNs on axillary US.

Answer> Thank you good comments. We had fine needle aspiration of suspicious LN on axillary US, however we did not investigate the number of suspicious LNs on axillary US additionally. We revised method in the revised manuscript.

Round 2

Reviewer 1 Report

This is a very interesting paper and should be further edited with a view towards publication, I really like this paper.  The abstract needs to be greatly simplified and I note the paper has been re-edited and is much improved but is not yet of a standard that is suitable for publication.